# Recent Technologies, Security Countermeasure and Ongoing Challenges of Industrial Internet of Things (IIoT): A Survey

**DOI:** 10.3390/s21196647

**Published:** 2021-10-06

**Authors:** Soo Fun Tan, Azman Samsudin

**Affiliations:** 1Faculty of Computing and Informatics, Universiti Malaysia Sabah, Kota Kinabalu 88400, Sabah, Malaysia; 2School of Computer Science, Universiti Sains Malaysia, Gelugor 11900, Penang, Malaysia; azman.samsudin@usm.my

**Keywords:** Industrial Internet of Things (IIoT), IoT architecture and networks, security and trust

## Abstract

The inherent complexities of Industrial Internet of Things (IIoT) architecture make its security and privacy issues becoming critically challenging. Numerous surveys have been published to review IoT security issues and challenges. The studies gave a general overview of IIoT security threats or a detailed analysis that explicitly focuses on specific technologies. However, recent studies fail to analyze the gap between security requirements of these technologies and their deployed countermeasure in the industry recently. Whether recent industry countermeasure is still adequate to address the security challenges of IIoT environment are questionable. This article presents a comprehensive survey of IIoT security and provides insight into today’s industry countermeasure, current research proposals and ongoing challenges. We classify IIoT technologies into the four-layer security architecture, examine the deployed countermeasure based on CIA+ security requirements, report the deficiencies of today’s countermeasure, and highlight the remaining open issues and challenges. As no single solution can fix the entire IIoT ecosystem, IIoT security architecture with a higher abstraction level using the bottom-up approach is needed. Moving towards a data-centric approach that assures data protection whenever and wherever it goes could potentially solve the challenges of industry deployment.

## 1. Introduction

The emergence of the Industrial Internet of Things (IIoT) acts as a new network paradigm that has transformed traditional capturing, collecting, exchanging, processing, and storing data in the industry. IIoT goes beyond the typical consumer devices, people-to-people (P2P) and people-to-machine (P2M) communication networks associated with the IIoT. IIoT consists of billions of “things” intelligently connected via distributed communication networks, such as machine-to-machine (M2M) communication. These “things” ranging from ultra-efficient sensors and actuators, automation devices, embedded systems, heavy machines to high-performance gateways, with real-time data analytics always present.

In most cases, these “things” are uniquely identified by a variety of addressing schemes, includes electronic product code (EPC), ubiquitous code (ucode) and media access control (MAC) and Internet protocol (IP) address. IIoT promises a transformative future for businesses and governments, including intelligent automation, smart factories, intelligent healthcare, smart homes, smart cities, and intelligent transportation. IIoT’s inherent complexities introduce several security challenges and privacy risks. Several surveys and reviews on analyzing IoT and IIoT security threats and privacy challenges have been published over the last decade. These existing reviews and surveys are chronologically summarised in Table 1.

In 2010, Atzori et al. [1] and Weber [2] initiated the studies of IoT security issues. Atzori et al. [1] briefly discuss IoT’s security challenges and privacy issues, particularly in RFID and WSNs. Weber [2] focuses on the security requirements, privacy legislation and personal data protection of the IoT and RFID. Miorandi et al. [3] provided an overview of IoT’s data confidentiality, privacy, and trust issues. Subsequently, Ziegeldorf et al. [4] gave a detailed discussion on privacy threats and challenges of IoT. Zhao and Ge [5] discussed security issues from the IoT architecture perspective and divided IoT into perception, transport, and application layers. Then, Jing et al. [6] further conducted a comprehensive analysis of each layer’s features, security issues, and corresponding solutions. After that, the discussion of IoT security is nailed down on the specific technologies and scope. The study of Fremantle and Scott [7] focuses the analysis on the middleware of IoT security. Granjal et al. [8] centralized on the security of IoT communication protocols, includes physical and medium access control (MAC) layers, IPv6 over low power wireless personal area network (6LoWPAN), routing protocol for low power and lossy networks (RPL). Nguyen et al. [9] focus on the security of IoT and WSN communication protocols and their attack-resistant solutions. Subsequently, Airehrour et al. [10] gave a detailed security analysis of IoT routing protocols, particularly in low-power and lossy networks (LLN). Then, Qin et al. [11] briefly discussed IoT security from a data-centric perspective. Loi et al. [12] directed to analyze consumer IoT devices. Fernández-Caramés et al. [13] and Lao et al. [18] review the adaptability of blockchain in securing IoT applications and architecture. Hassija et al. [14] focus on discussing the security of IoT applications. Berkay et al. [15] and Tabrizi and Pattabiraman [16] directed to review the IoT security from a programming platform and code-level perspective. Amanullah et al. [17] discuss the relationship between deep learning, IoT security and big data technologies. Joao et al. [19] gave a general review of threat models and attack paths of IoT.

Recent IIoT surveys have primarily focused on the general IoT domain rather than the IIoT domain. They either provided a general overview of IoT security [1,2,3,4,5,10,11,19], or a detailed security analysis limited to specific IoT technologies or a particular layer of IoT architecture [6,7,8,9,12,15,16]. In addition, multiple surveys focused on exploring the relationship between IoT security and blockchain technologies [13,14,17,18]. Survey directions have lately been directed to be hammered down in the IIoT domain [20,21,22,23,24]. Deep learning in IIoT threat detection [20,22] and decentralised blockchain technologies [22,23] are the focus of these IIoT security surveys. However, none of them performs comprehensive security analysis on IIoT architecture and its recent industry solutions. Whether these deployed security solutions in the industry are still adequate to be adapted to secure IIoT architecture are questionable. The contributions of this article are: The difference between conventional systems and IIoT security concerns are summarized. Decentralized security approaches with high scalability, high interoperability, lightweight, and secure data processing have urged to address the high heterogeneity of “things,” high volume, and variety of collected sensor data, as opposed to conventional security systems focused on a centralized approach.Unlike recent IIoT architectures [24,25,26,27] that (i) focused on specific industries: aviation industry [25] and smart manufacturing [27], and (ii) targeted on particular technologies: M2M communication [24], green-aware multi-task scheduling [26] and 5G technology [27], we generalized the IIoT architecture into a four-layer architecture to cope with a wide of industry technologies and standards.Subsequently, we classify the recent IIoT technologies and standards into the proposed four-layer IIoT architectureThe IIoT security requirements are further defined with the CIA+ model, includes confidentially(C), integrity(I), authentication(A), authorization and access control (A) and availability (A).A comprehensive end-to-end security analysis was conducted based on the defined IIoT CIA+ model. Subsequently, a fine-grained review on recent industry technologies and standards in each layer of the proposed IIoT architecture. The identified security risks and threats of these industry technologies, their deployed security countermeasures and future research works are summarizedLastly, we enumerate the open security challenges of IIoT and future research opportunities.

The rest of this article is organized as follows. Section 2 investigate the characteristic of IIoT, highlights and report the difference between conventional systems and IIoT security concerns. Section 3 review the recent works of IIoT architecture and propose an IIoT security architecture based on the ITU-T Y.2060 IoT reference model [20], consisting of four layers: device layer, transport and network layer, processing layer and application layer. Then, we classify the recent industry technologies and standards into the proposed IIoT security architecture. Subsequently, Section 4 presents a comprehensive end-to-end security analysis on each layer of IIoT architecture by using the CIA+ model. The security risks and threats of each industry technology and their deployed security countermeasure, the gaps of today’s deficiency, and ongoing challenges are reported. Section 5 discusses the open security challenges, privacy issues and future research opportunities of IIoT. Finally, Section 6 concludes.

## 2. IIoT Security Challenges and Concerns

The discussion of IIoT can be traced back to the connection between the physical world and ubiquitous “things” via the Internet during the early 1990s [28]. While IIoT was still in its infancy growth stage, these definitions’ scope is framed by different business interests and industry application scenarios [29,30,31,32,33]. For example, IETF and IEEE definitions are bounded by sensing technologies such as RFID and sensors [29,30], whilst the W3C expound the IoT with the Word Wide Web ecosystems [31]. IoT’s vision is to enable the connection of any “things” anytime. In most industry cases, we concluded that these “things” are associated with three fundamental characteristics: heterogeneity, unique identities and connectivity.

Along with the growth of IIoT for supporting industries, IIoT security and privacy issues have become more challenging. These security challenges inherit the conventional systems issues such as the advanced persistent threat (APT) and are further exacerbated by the complexity of the newer IIoT associated characteristics such as high heterogeneity, large scale of “things”, and cyber-physical systems. Table 2 further summarises the difference between conventional systems and IIoT security concerns.

The high heterogeneity of “things” on a large scale implicates the interoperability issues of cross-network communications, cyber-physical systems and IIoT enabled-technologies integration. The intricate maze of interoperability issues arises when: (i) heterogeneous devices and sensor nodes are identified with different naming and addressing schemes; (ii) exploit different data structures and formats; and (iii) communicate through different security protocols with varying requirements of the network (e.g., reliability, communication cost, latency and bandwidth) and integrated to provide a plethora of service applications. The question of whether these conventional security mechanisms and defence systems can be further integrated and standardized universally in resolving IIoT security complexities remains unanswered.

When there is a large scale of “things” (e.g., sensors in the aviation industry that consistently capture engine and aircraft health information during a flight) or diverse “things” in smart factories and manufacturing (e.g., sensors, edge devices, and smart grid) that collaborate to generate and exchange data continuously, these generated data from cyber-physical systems always come in big data flavour [17]. The data come in high volume and wide variety (e.g., structured, unstructured, quasi-structured, and semi-structured data), which need to be processed at a high velocity or analyzed nearly real-time, resulting in conventional data processing mechanisms being complicated or too expensive to scale and handle them efficiently.

As conventional data processing systems mainly were built-in houses, centralized management, and typically worked within the organization boundaries with a finite number of connected devices and users; therefore, security and privacy issues were not a concern. However, security protection and defences mechanisms are significantly different in the era of IIoT. Collected sensors data are locally processed and analyzed by IIoT gateway or automation system before sending to a centralized cloud platform for remote monitoring and post-analysis. The scalability of the existing security mechanisms to authenticate, fine-grained access control on massive IIoT resources has drawn the industry and researcher’s attention to move forward into a decentralized approach. Subsequently, more lightweight and highly efficient encryption schemes have been proposed recently to protect the tiniest “things” of IIoT, such as edge devices, sensor nodes and WSNs.

## 3. IIoT Architecture

### 3.1. Overview of IoT and IIoT Architecture

The origins of IIoT architecture can be traced back to the early designs of IoT architecture. In 2011, Ning and Wang [34] proposed a future IoT architecture called a U2IoT model. The U2IoT architecture works similar to a human nervous system that consists of unit IoT and ubiquitous IoT. The unit IoT serves as a local unit based on the man-like nervous model and is responsible for handling and managing diverse local IoTs. The ubiquitous IIoT follows the blueprint of social organization framework architecture and is responsible for integrating, managing, and controlling the collaboration among multiple IoT units across the industry, nationwide and worldwide. On the other hand, Guinard [35] worked on the concepts of web of things (WoT) by proposing an architecture that integrates the connection of “things” to the existing web services via existing web technologies. The proposed WoT architecture consists of five layers, includes accessibility, findability, sharing, composition and application layers. Subsequently, Gomez and Lopez [36] extended the WoT concepts into a hybrid distributed IoT architecture that consists of two distinct resource-oriented approaches: WoT and Tripe Space. WoT underlying a hypertext transfer protocol (HTTP) to interconnect the IoTs in the world wide web. Tripe Space applies semantic web protocol to exchange machine-processable data among the heterogeneous devices in the distributed local shared space. Vernet et al. [37] further customized the WoT architecture into the Smart Grid domain. Meanwhile, Olivier et al. [38] and Qin et al. [39] proposed another IoT architecture based on software defined networks (SDN) that consists of three layers: infrastructure layer with interconnecting network devices; control layer that comprises of SDN controllers; and an application layer that includes the applications for configuring the SDN. On the other hand, several research projects such as IoT-A [40], iCore [41], Sensei [42], and COMPOSE [43] have proposed a reference architecture of IoT at a high abstraction level.

A step closer to real-world industry implementation, several researchers [44,45,46,47,48] and vendors (i.e., Finnode, ThingWorx and Xively) use cloud technologies to tackle the IIoT heterogeneity issues and scalability services. These cloud-centric IIoT architectures use a centralized or decentralized cloud platform to process and manage the aggregated data from heterogeneous networks such as RFID, WSN, and body area network (BAN). These cloud-based IIoT platforms also provide API interfaces for industries to develop their IIoT applications. Researchers [49,50,51] recently attempted to integrate blockchain technologies in solving the decentralized issues of cloud-based IIoT architecture. Whether these blockchain-based architectures are practicable to support a large scale of things with their constrained resources in real-world industry implementation needs to be further investigated.

Generally, the initial widely accepted IIoT architecture is constructed based on the three-layer architecture [6,52], namely the perception, network, and application layers. The perception layer consists of the “things” identification and sensing technologies to collect and exchange the data. The network layer enables the communication and data transmission between the perception layer and the application layer. In most cases, it also involves data aggregation and curation process. Lastly, the application layer confluxes the data aggregated and virtualises the analysed result based on society, business and government demands. Different business interests reflect various IIoT applications for this layer, such as smart cities, intelligent health and smart transport. As three-layer architecture confronted the interoperability and scalability problem to well-suit into existing Internet and telecommunication networks, Wu et al. [53] extended the three-layer architecture into five-layer architecture by proposing a new business layer that resides on the top of the application layer and further dividing the previous network layer into processing layer and transport layer. The transport layer is responsible for transmitting the data generated from the perception layer into the processing layer. The processing layer focuses on processing, storing, and performing analytical works based on the application layer’s demand. While the application layer consists of diverse IIoT applications customized to each industry requirement, the business layer monitors these applications’ release and charging, conducts research on business and profit models, and controls privacy issues. Subsequently, ITU [52] proposed an IIoT reference architecture that consists of four layers: device layer, network layer, service and application support layer and application layer. The device layer is responsible for capturing and uploading data directly or indirectly via communication networks or gateway protocol, such as controller area network (CAN) bus, ZigBee and Bluetooth. The network layer is capable of handling network and transport connectivity. The service and application support layer aimed to provide a support function for various IIoT applications includes data curation, processing or storage. The application layer consists of IIoT applications. Thereafter, Cisco [54] proposed a seven-layer IIoT reference architecture comprising physical devices and controllers, connectivity, edge or fog computing, data accumulation, data abstraction, application, collaboration and processes layers. The physical devices and controllers layer includes various endpoints that can generate data, be queried and managed. The connectivity layer refers to the communication and connectivity either between devices, local networks or across the networks globally. Transforming network data flows into an appropriate data format for high-level data processing and storage occurred in the edge or fog computing layer. The data accumulation layer is responsible for data storage, whereas the abstraction layer involves aggregating and rendering data and storage to serve the client application. The application layer refers to the IIoT application such as business intelligence and big data analytic applications, sensors control applications and mobile applications. The relationship between the three-layer, four-layer, five-layer and seven-layer IIoT architectures is correlated, and their correlation is further mapped and illustrated in Figure 1.

### 3.2. The Proposed IIoT Security Architecture

This subsection presents the proposed four-layer IIoT security architecture, as illustrated in Figure 2. We propose a four-layer IIoT security architecture to solve the shortcomings in current IIoT architectures [15,22,23,24,25,26,27,28,29,30,31,32,33,34,35,36,37,38,39,40,41], which are generic and difficult to address in industrial settings. For example, three-layer IoT architecture fails to satisfy the need for data curation, processing, and storage in IIoT. Subsequently, we classify recent industry IoT technologies and standards into the proposed IoT security architecture for conducting end-to-end security analysis, and the results are further discussed in Section 4. The security analysis on the device layer focuses on the physical and virtual “things” identification schemes used to connect to IIoT networks. These schemes include EPC, ucode, MAC and IP addresses. On the other hand, security analysis on transport and network layers focuses on IIoT communication technologies and standards, including capillary, backhaul, and backbone networks. The processing layer addresses the end-to-end data protection issues of IIoT data processing platform. Lastly, the application layer addresses the application threats, host-to-host, and client-server application protocol challenges, such as simple object access protocol (SOAP), representational state transfer hypertext transfer protocol (REST HTTP) and data distribution service for real-time systems (DDS).

## 4. End to End Security Analysis on the Proposed IIoT Security Architecture 

Security always serves as a linchpin of the public adoption of the new technologies. Any deficiencies of the security protection and defences system of IIoT has a latent risk to decelerate its adoption. This section conducts a comprehensive end-to-end security analysis based on the proposed IIoT security architecture. The security analysis is conducted based on the CIA+ model [7,55,56] described in Table 3.

### 4.1. Device Layer

The device layer consists of a large scale of “things” distributed across the IIoT infrastructure landscape. These heterogeneous “things” need to be identified uniquely before being interconnected and collaborate to capture, transfer, exchange and process data. Generally, these identification schemes and technologies must be unique, consistent, persistent and able to support the identity management and scalability issues of “things” [57,58]. Recently, several naming and address schemes have been used to define the unique identifier for both the physical “things” and virtual “things”, either within a local or global scope. These schemes are electronic product code (EPC), ubiquitous code (ucode), media access control (MAC) and Internet protocol (IP) addresses, and other higher layer identifiers and naming schemes such as uniform resource name (URN), object identifier (OID), digital object identifier (DOI), network basic input/output system (NetBIOS), etc.

#### 4.1.1. Electronic Product Code (EPC)

Auto-ID Labs, MIT developed electronic product code (EPC), and it is widely used today to issue a unique 64-bit or 96-bit globally unique identifier (GUID) for physical “things” [59,60]. In most cases, EPC uses RFID tags to identify “things”, although it can also be used with optical data carriers, including linear bar codes, 2D barcodes, and data matrix symbols. Header, EPC manager number, object class, and serial number are the four components of an EPC identifier. The header identifies the length, structure, type, version and generation of the EPC. The EPC manager number is an entity responsible for maintaining the subsequent partitions. In most case refers to the manufacturer or company that produces the product and responsible for attaching the EPC. The object class identities a class of objects. The object is a product type, and it most likely refers to the stock keeping unit (SKU) [59,60].

Meanwhile, the object name service (ONS) is responsible for handling EPC information lookup services. However, ONS is established under the domain name system and lacks an authorization and authentication mechanism for handling ONS queries. The detailed security analysis of the EPC and its ONS lookup service are summarised in Table 4. Overall, the enforcement of more robust security properties (e.g., MD5 and SHA-1 hashing algorithms) on EPC is currently limited by its very constrained storage and computation power, i.e., less than 1000 bits in EPC Gen2 tags [60]. Subsequently, the scalability and extendibility of ONS and EPC networks issues need to be addressed to support the IIoT applications.

#### 4.1.2. Ubiquitous Code (Ucode)

Ubiquitous code (ucode) was developed by Japan to uniquely identify objects, places, and concepts in the real world with a length of 128-bit. In the future, it can be further extended to the multiple of 128-bit, includes 256-bit, 384-bit, and 512-bit [70,71,72]. The ucode consists of five components: version, top-level domain code, class code, second-level domain code, and identification code. The ubiquitous ID center is responsible for establishing the ucode standard and its ubiquitous ID architecture consisting of ucode, ucode tag, communicator, resolution server, and information server. The recent ubiquitous ID architecture is secured with eTRON security framework, and its detailed analysis is presented in Table 5. Overall, security enforcement and privacy protection mechanisms exist on the ucode and its Ubiquitous ID architecture. Recently, the Ubiquitous ID Center is moving forward the ucode and its architecture to support IIoT’s scalability and interoperability issues, such as designing the hierarchical and distributed resolution server and information server.

#### 4.1.3. Media Access Control (MAC) and Internet Protocol (IP) Address

Media access control address (MAC), also known as physical address, hardware address, ethernet hardware address (EHA) and adapter address, is a unique identifier of the physical or virtual network node. These include network interface cards, firmware devices, hardware and software devices. The MAC address naming scheme is managed by IEEE. Recently, it follows three naming spaces standards, includes MAC-48 (e.g., Ethernet, Bluetooth, IEEE 802.11 wireless network, IEEE 802.5 token ring and ATM), EUI-48 (intended to replace MAC-48 and used for IEEE 802-based network applications) and EUI-64 (e.g., IPv6, Zigbee, 6LoWPAN and IEEE 802.15.4) [75]. Generally, MAC address consists of a 24-bit organization unique identifier (OUI) assigned by the IEEE registration authority and a 24-bit to 40-bit extension identifier assigned by the OUIs’ manufacturer.

Meanwhile, the Internet protocol (IP) address is a unique identifier that relies on Internet Protocol for network access and communication. Internet Protocol version 4 (IPv4) is a 32-bit identifier that is widely deployed today. However, with the emergence of IIoT and the dramatic growth of network-enabled devices, the IPv6 address is used to solve the root problem of IPv4 address exhaustion instead of using trivial solutions such as network address translation (NAT), virtual or private network addresses. IPv6 is a 128-bit identifier that consists of six parts: 3-bit format prefix (00,1 FP), 13-bit top-level aggregation identifier (TLA ID), 8-bit future use reservation (RES), 24-bit next-level aggregation identifier (NLA ID), 16-bit site-level aggregation identifier (SLA ID), and 64-bit interface identifier. The IP address focuses on network layer communication located at Layer 3 of the open system interconnection (OSI) model. The MAC address aims to transmit the data link layer or Layer 2 of the OSI model. Both IP address and MAC address can work together with address resolution protocol (ARP).

MAC and IP addresses still suffer from numerous security threats and privacy issues, as summarised in Table 6. Whether the convention security mechanisms (e.g., cryptographic algorithm, firewall, intrusion detection and prevention system) and recent IPsec security mechanism (e.g., authentication header (AH) protocol, encapsulation security payload (ESP), etc.) are sufficient to protect their confidentiality, authentication, and integrity is still questionable with the recent widespread of advanced persistent threat (APT) attacks. Moreover, IPsec only serves as a security standard, and there exists an interoperability issue of implementing secure communication protocol across diverse manufacturers. For instance, HP does not support the Diffie-Hellman Group 1 key on Internet key exchange (IKE).

### 4.2. Transport Layer

Transport Layer mainly provides ubiquitous connectivity for “things” and transmitting generated data from the device layer to the processing layer via a heterogeneous collection of communication networks, as illustrated in Figure 2. This section analyses the security of the transport layer based on its range of coverage and functional architecture, mainly capillary network, backhaul network and backbone network.

#### 4.2.1. Capillary Network and Communication Technologies

The capillary network is defined as a local network that is intelligently connecting “things” (e.g., sensors, actuators, embedded devices) via short-range radio access, power link communication or infrared technologies [79,80]. These communication technologies include IrDA, Bluetooth, radio frequency identification (RFID), near field communication (NFC), INSTEON, Bluetooth, Bluetooth low energy (BLE), EnOcean, ultra-wideband (UWB), ANT+, HomePlug, ZigBee, WirelessHART, ISA110.11a and Thread [79,80,81,82,83,84], as illustrated in Figure 2. The detailed security analysis of these technologies is further summarised in Table 7.

Generally, most capillary communication technologies are radio waves-based and wireless, except for IrDA and HomePlug, which use infrared light and power link communication. These wireless and radio waves-based communication technologies are generally vulnerable to wireless network threats, man-in-the-middle attacks, unauthorized message resource misappropriation. Meanwhile, the Bluetooth, BLE, WirelessHART and Thread technologies, exposure a higher security risk operating on popular radio frequency—2.4 GHz ISM band. Although most of them follow the IEEE 802.11 security standard, however, still subjected to a broad-spectrum jamming attack, 802.11 frame injection, 802.11 data replay, 802.11 Beacon, and authenticate flooding attack. Compared to others, UWB enjoys more robust security features as it operates on very low radiated power and narrow pulses, which increases wireless attack difficulties.

At the edge of IIoT networks, the ongoing security challenges such as interoperability issues of different security mechanisms (e.g., incompatibility of AES encryption in multichannel mode), scalability of key management and distribution, stronger and lightweight cryptography algorithm in the constrained resource of “things (low power, energy, storage size) and end-to-end communication and data protection across the heterogeneous interface are urgently called for a realization of a complete IIoT vision.

#### 4.2.2. Backhaul Network and Communication Technologies

The backhaul network, which sits between the capillary and backbone networks, controls circulating data packets. Backhaul network also serves as a bridge in the heterogeneous capillary communication technologies. This section focuses on the analysis of the backhaul networks, which includes ethernet, wireless local area network (WLAN, or Wi-Fi), low rate-wireless personal area network (LR-WPAN), WiMax and universal mobile telecommunications system third generation (UMTS 3G) [96,97,98,99,100,101], as presented in Table 8.

Overall, these communication technologies satisfy the network security requirements, includes the physical and MAC layers security protection, message confidentiality and integrity check, authentication and authorization. However, with limited availability in protection, the strength of their defences countermeasure is varied and strictly dependent on their supported cryptographic algorithms and physical environments, includes the network range and coverage, i.e., WLAN enjoys lower network attack risk compared to WiMax.

As most backhaul communication technologies can be applied to support capillary communication (e.g., LR-WPAN) and backbone network (e.g., UMTS 3G and WiMax) in the domain, their role here is more focused on building a bridge or gateway for the data across heterogeneous capillary and backbone networks. Several security challenges need to be further addressed. These include: (i) end-to-end security protection over the bridge between capillary network and backbone networks; (ii) scalability of network size and bandwidth to support the rapid growth of IIoT data packets; (iii) interoperability with existing capillary communication technologies; (iv) stronger but lightweight security mechanisms on a constrained network resource; (v) more sophisticated access control scheme, such as identity-based or attributed-based access control; and (vi) efficient network key management and distribution.

Several low power and energy consumption networks have been developed recently to address the resource-constrained issues of IIoT. These works include low power wireless personal area network (6LoWPAN) and Wi-Fi Halow. Internet Engineering Task Force (IETF) has established a new protocol called IPv6 over low power wireless personal area network (6LoWPAN) to facilitate the communication between 802.15.4 enabled devices and the Internet. With header compression and address translation technique, IPv6 data packets are encapsulated into 802.15.4 data packets, thus enabling the integration of the 802.15.4 based network with the IPv6 network. Meanwhile, Wi-Fi alliance announced a new extension of Wi-Fi standard–802.11 ah, also known as WiFi-Halow. WiFi-Halow provides lower energy connectivity and a more extended range than traditional Wi-Fi, thus supporting a large scale of sensor stations or nodes.

#### 4.2.3. Backbone Networks 

The backbone network is a core network capable of interconnecting various backhaul networks and providing various services or gateways to facilitate communication and information exchange. In some cases, the backbone network may directly connect to a capillary network without a backhaul network. Backbone communication technologies consist of a wired network (IEEE 80.3 Ethernet), wireless network (IEEE 802.11, IEEE 802.16) and cellular network (2G, 3G, LTE, 4G and 5G). This section analysis the backbone communication technologies in the scope of IIoT long-range wide area network (LR-WAN) that includes NarrowBand-IoT (NB-IoT), long range WAN(LoRaWAN), narrowband fidelity (NB-Fi), weightless, SigFox and Ingenu random phase multiple access (RPMA) [101,102,103,104,105,106,107]. The detailed security analysis of each technology is summarised in Table 9.

Most of these technologies use cloud technologies to support their network architecture, including LoRaWAN, NB-Fi, SigFox and DASH7. The gateway or base station is responsible for forwarding the received “things” data packet to a cloud-based network via backhaul networks. The majority of these LR-WAN technologies are found to have general security properties such as using the AES-128 bit for data confidentiality, password-based authentication and access control and unique device identity [80].

### 4.3. Data Processing Layer 

Data processing layer is responsible for transforming network data flows into valuable information (e.g., data pre-processing and curation), subsequently storing in a data warehouse and using high-level data processing such as aggregating, analyzing and interpreting data to serve client applications. In most cases, IIoT data comes with the Big Data characteristic when there is a large scale of “things” generate and exchange data continuously, such as in smart cities, aviation industry, logistic and shipment industry. IIoT technologies offer automated mechanisms such as cloud, edge and fog computing, machine-to-machine (M2M) communication technologies and underlying architecture to capture, collect and transmit data into the warehouse. On the other hand, big data technologies provide a data processing platform to support real-time analytics, such as Hadoop in a data warehouse to curate, process and analyze these machine data and turn it into valuable information. Because of the technological challenges of implementing in-house big data processing, these data are frequently stored and processed by third-party service providers such as Hortonworks, AmazonEMR, Cloudera, IBM, Zettaset, HDInsight, etc. As a result, the issues of data are exacerbated [108]. This section discusses end-to-end data protection for the data processing layer.
Data-in-TransitData-in-transit refers to data being transmitted from the network layer to the data processing layer and application layer, either forwardly or backwardly. Most of big data technologies relies on Kerberos authentication scheme, public key infrastructure (PKI) and network encryption algorithm such as Hadoop remote procedure call (RPC), secure socket layer (SSL), HDFS data transfer protocol, simple authentication and security layer (SASL) mechanism, to ensure data confidentiality and authentication. However, these security mechanisms provide limited access control and authorization capabilities. A more sophisticated access control scheme such as role-based, identity-based and attribute-based can be plug-in with additional security packages such as Cloudera Sentry, DataGuise, IBM Infosphere Optima Data Masking, Datastax Enterprise, Zettaset Secure and others [108,109]. While data-in-transit protection in IIoT is mainly constructed based on the conventional security mechanism such as PKI, it is recommended to employ the bottom-up approach for heterogeneous and distributed networks infrastructure. The more challenging issue here is the key management and distribution problem, includes distribute key across distributed network and communication technologies, maintaining a large scale of the certificate, key revocation, recovery, and updating process. The key management proposal should be able to solve the scalability and interoperability issues to solve these problems.Data-at-RestData-at-rest refers to the data being stored in persistent storage such as a disk file. Conventional data-at-rest protection approaches include installing tamper-resistant hardware in third-party service providers, full-disk encryption, database-level encryption, table level encryption, and application-level encryption. Data are encrypted in the application layer before being inserted into the database. For instance, TrustDB and CipherBase provide data-at-rest protection based on the co-design of hardware and software [110].Data-in-TransformData-in-transform indicates that data is subjected to various means and manipulation methods, including performing query, sorting, mathematical operations, statistical analysis, and other functions on data to produce meaningful output. Protection of data-in-transform in IIoT is critically vital as their necessities in supporting real-time data analytics. Most of the recent big data processing service providers are still inadequate to support confidentiality during data transformation.

While conventional security mechanisms are limited to protect data-at-rest and data-in-transmit, the recent advancement of homomorphic encryption algorithms can be used to ensure the security of data-in-transform. Homomorphic encryption is a data encryption algorithm that works similar to a conventional data encryption algorithm, however, with the added capability to perform computation over encrypted data. Therefore, homomorphic encryption can serve as a comprehensive data protection solution to protect real-time analytics in IIoT. Recently, homomorphic encryption has been commercialized and released. Exiting products include CryptDB, MrCrypt, Crypsis and computing on masked data (CMD). CryptDB focuses on protecting MySQL query, MrCrypt supports MapReduce operation, Crypsis aimed for high-level data flow language such as Pig Latin, and CMD protecting NoSQL environment [109]. However, all of them are constructed based on a partial homomorphic encryption scheme, which can support simple arithmetic operation either additive or multiplicative homomorphism, still inadequate to enable artificial intelligence (AI) capabilities and machine learning algorithms. Most of these solutions lead to an increased data storage size (approximately 3.76 times in CryptDB), communication cost and bandwidth. A more efficient homomorphic encryption algorithm is needed to provide end-to-end protection for real-time data analytics in the IIoT data processing layer.

### 4.4. Application Layer 

The application layer is a high abstraction level that leverages transport layer protocol, such as transmission control protocol (TCP) and user datagram protocol (UDP), to support data transfer and exchange between host-to-host client-server or peer-to-to-peer model. This section focuses on IIoT related session layer protocols includes Simple object access protocol (SOAP), representational state transfer hypertext transfer protocol (REST HTTP), data distribution service for real-time systems (DDS), message queue telemetry transport (MQTT), extensible messaging and presence protocol (XMPP), advanced message queuing protocol (AMQP), and constrained application protocol (CoAP) [111,112,113,114]. The details of their security analysis are presented in Table 10.

Message exchange pattern (MEP) of these protocols can be categorized into two groups, request/response pattern (SOAP, REST HTTP, XMPP, CoAP) and subscribe/publish pattern (DDS, MQTT, XMPP, AQMP). Request/response pattern is the most commonly used synchronous pattern, in which the client requests information from the service by sending a request message and expects a response message from the service within a defined timeout. Subscribe/publish pattern is an asynchronous pattern that broadcasts data regularly to subscribers interested in their data, providing better network scalability and greater dynamic network topology.

Security mechanisms of these protocols are primarily dependent on transport layer security/secure socket layer (TLS/SSL) mechanism. Communication and data protection of MQTT, XMPP and AMQP are directly secured by using TLS/SSL. The communication of REST over an HTTP is encrypted with a TLS/SSL connection. The datagram transport layer security (DTLS) in CoAP is a stream-oriented TLS/SSL and inherits most of the features from TLS/SSL without any optimization for a constrained resource environment. SOAP enjoys a stronger security feature with its web service security (WS-Security). However, it spends much bandwidth in communicating metadata.

As TLS/SSL mechanism only provides limited security assurance for confidentiality, integrity and authentication, these protocols are still vulnerable to application layer threats (e.g., SQL injection, invalidated object indirect reference, DoS Attack and cross-site request forgery), SSL stripping, and password-based attacks (e.g., brute force attack, dictionary attack and rainbow table attack). Conventional cryptographic techniques such as MD5, DES, SHA-1, and RSA are being used to bolster their security defences, despite the fact that these algorithms have been broken and demonstrated to be insecure. In the future, several issues such as lightweight cryptography, reliability of message communication, interoperability issues with heterogeneous nodes, and the scalability of public key infrastructure to handle a large scale of key management need to be addressed for further spurring IIoT development.

## 5. Open Security Issues and Privacy of IIoT 

IIoT is not a single technology; in turn, it leverages on various existing technologies such as sensing technologies, communication networks, high-performance processing platforms (e.g., cloud computing, M2M and edge computing), and also the new emerging technologies (e.g., LoRaWAN, NB-Fi and NB-IoT) to form its entire ecosystem. Consequently, IIoT security and privacy concerns are not merely focused on the issues of single piece technology. It encompasses a whole range of IIoT ecosystems, ranging from the physical security of connected nodes or devices, communication security of networks, data security during the transmission, transformation and storage of data to IIoT application security. This integrated heterogeneous environment becomes critically challenging with the employment of diverse security protocols, defence mechanisms, and standards across the IIoT architecture. Most of them employ conventional security approaches to build up their defences and protection systems for securing data and communication. Whether these deployed conventional mechanisms are still adequate to protect recent IIoT technologies is questionable, this section discusses IIoT’s open security and privacy issues.
Security architecture and framework for IIoT.As the IIoT is still in its early stages of development, distinctive security models and designs were proposed as of late to address its security challenges and privacy concern issues. The point of these works are focused to make sure about a particular: (i) IIoT architecture, includes cyber-physical social based security model [109] focused on the U2IoT architecture [34] and Grid of security approach [38] targeted for securing SDN-based IIoT architecture; (ii) “things”, i.e., OSCAR [115] directed to protect constrained application protocol (CoAP) communication networks, Physically Unclonable Functions (PUFs) based authentication protocol [116] for ensuring RFID framework; (iii) systems, includes a lightweight security system focused for IPsec, DTLS, and IEEE 802.15.4 connection layer. As no single security architecture and framework can fix the entire IIoT ecosystem, a design of IIoT security architecture with a higher abstraction level by using the bottom-up approach is needed. The proposal’s emphasis should be on interoperability issues to integrate different security mechanisms supported by IIoT technologies and cross-layer security solutions. Below are some of the highlighted issues in this domain. How can encrypted data be passed through different network layers from the physical layer, transport layer to application layer securely supported by different communication technologies? How can the things identified with a different addressing scheme secured under different security mechanisms communicate universally? How to exchange data securely with a different set of data formats (e.g., XML, JSON, etc.)? How to implement a secure communication protocol across a diverse manufacturer? Besides that, scalability issues should be further addressed to support rapid growth in IIoT, such as the scalability of PKI to manage a large scale of X.509 certificates.Limitation of conventional point-to-point defenses system and security mechanisms.The connection and communication across IIoT networks are recently protected via conventional network security protocols such as TLS/SSL, IPSec, RADIUS, IKE, etc. Most of these security protocols work based on point-to-point defences. For instance, TLS/SSL offers protection over the transport layer, IPSec focuses on IPv6 and IPv4 MAC, data link, transport and network layer. As IIoT communication technologies diversify, these conventional security mechanisms that focus on point-to-point defences are less efficient against the new cyber advanced persistent threats (APT) attacks and malicious insider attacks. The malicious attacks can target any vulnerabilities or weak points of IIoT networks or application systems. For instance, multi-hop wireless broadcast communication is vulnerable to eavesdropping. These situations become worst with the bring your own device (BYOD) or bring your own technology (BYOT) environments. The hijacked or backdoor installed devices can penetrate IIoT networks easily.Lightweight and stronger cryptographic algorithmMost IIoT communication protocols and technologies still rely on conventional cryptographic algorithms such as RSA, MD5, RC4, and DES-56 to ensure data confidentiality and secure communication. However, some of these algorithms have been proven insecure and subjected to quantum attacks. Therefore, a more robust cryptographic algorithm needed to be adapted into these communication protocols, such as quantum-resistant NTRU and BLISS algorithms. Besides that, a lightweight but secure algorithm is highly sought after to protect the IIoT constrained resources (e.g., low energy, low storage and low bandwidth communication). For instance, RC5, SkipJack, high security and lightweight (HIGHT), corrected block TEA (XXTEA), SAFER++ have been proposed recently to secure wireless sensor networks [117,118].Limitation of IPSec and TLS/SSL mechanismMost IIoT technologies rely on IPSec and TLS/SSL mechanisms to secure their communication and data transmission. Both provide confidentiality, integrity and authentication of the message, with limited authorization and access control. However, without the assurance of availability and non-repudiation, they are still vulnerable to application-layer threats. Besides that, the scalability and interoperability issues of both IPSec and TLS/SSL need to be addressed further. These include scalability of key management and distribution to handle a large of “things” network keys, the implementation issues in a lightweight and constrained resource protocol such as CoAP.Password-based authentication schemeMost IIoT authentication schemes are still constructed based on single-factor authentication (SFA)—user’s ID and password. Some network keys are further derived from the user’s password. However, these short, weak, easily predictable and repeated passwords further paring security defences mechanism and subjected to a brute-force attack, dictionary attack, rainbow attack, etc. Recently, the two-step verification mechanism can be activated optionally, in which four- or six-digit verification code will be sent to a user via SMS or voice call, or alternative can be retrieved from the time-based one time password apps.Towards a data-centric approach for end-to-end data protectionAs no single security mechanism and framework can fix the entire IIoT ecosystem due to its inherent, the data-centric approach can serve as another alternative towards end-to-end security for IIoT. Instead of targeting to protect different networks, communication technologies and protocols, the data-centric approach aims to protect the data itself—whenever and wherever it goes. These data-centric approaches include homomorphic encryption, attribute-based encryption scheme, private information retrieve scheme, searchable encryption scheme and multi-party computation scheme [119,120,121,122]. Most of these schemes can assure data-in-transit, data-in-transform and data-at-rest security, thus significantly resolving the interoperability and scalability issues of integrating different security mechanisms across IIoT networks and technologies. These schemes also significantly reduce the risk of privacy (e.g., collection and abuse use of personal data, habits and geolocation). Homomorphic encryption, for example, permits a third-party data processing centre to undertake real-time analytical work without having to decrypt data collected from any industry.

## 6. Conclusions

This article provides a comprehensive study of IIoT security architecture and associated industry technologies and standards. Firstly, this article discussed the IIoT definitions, IIoT characteristics and highlighted IIoT security concerns that are different from the existing data security concerns. Subsequently, the paper reviewed current IIoT architectures and proposed a new four-layer IIoT security architecture. The proposed security architecture is constructed based on a bottom-up approach. A comprehensive end-to-end security analysis on each layer of the proposed IIoT architecture is conducted. This includes assessed security requirements based on CIA+ model, highlighted their recent industry counter measurement and deficiencies, discussed ongoing security challenges and future works. The security analysis on the device layer focuses on physical and virtual “things” identification schemes, including EPC, ucode, MAC, and IP addresses. The analysis on transport and network layer is further sub-divided into the capillary network (IrDA, RFID, NFC, INSTEON, Bluetooth, BLE, EnOcean, UWB, ANT+, HomePlug, ZigBee, ISA 110.11a, WirelessHART and Thread), Backhaul network (Ethernet, WLAN, LR-WAN, WiMax, 3G) and backbone network (NB-IoT, LoRaWAN, NB-Fi, NWAVE, SigFox, RPMA and DASH7), based on their range of coverage and functionalities. Security analysis on the processing layer focuses on end-to-end data protection, includes data-in-transit, data-at-rest, and data-in-transform. Finally, the application layer studies focused on SOAP, REST HTTP, DDS, MQTT, XMPP, AMQP and CoAP protocol. This article also highlighted IIoT’s open security and privacy issues, including constructing a standardized IIoT security architecture in a bottom-up approach.

## Figures and Tables

**Figure 1 sensors-21-06647-f001:**
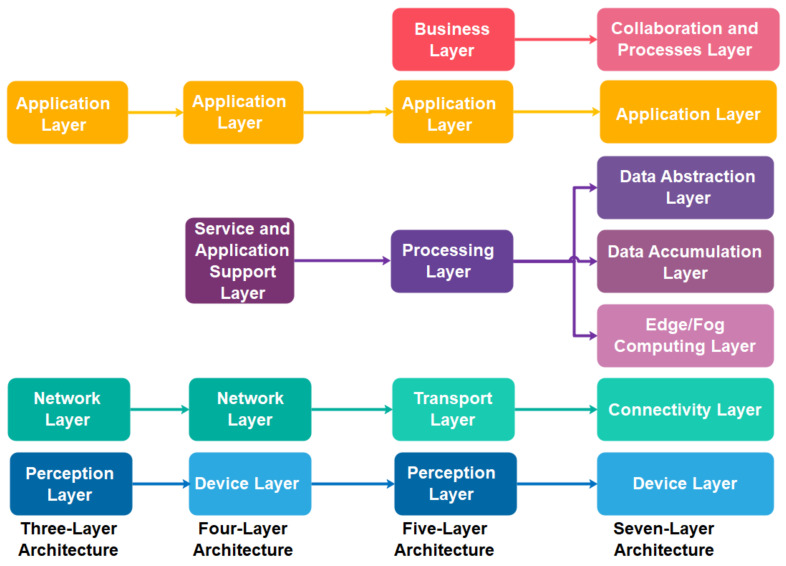
The relationship and mapping of three-layer, four-layer, five-layer and seven-layer IIoT architectures.

**Figure 2 sensors-21-06647-f002:**
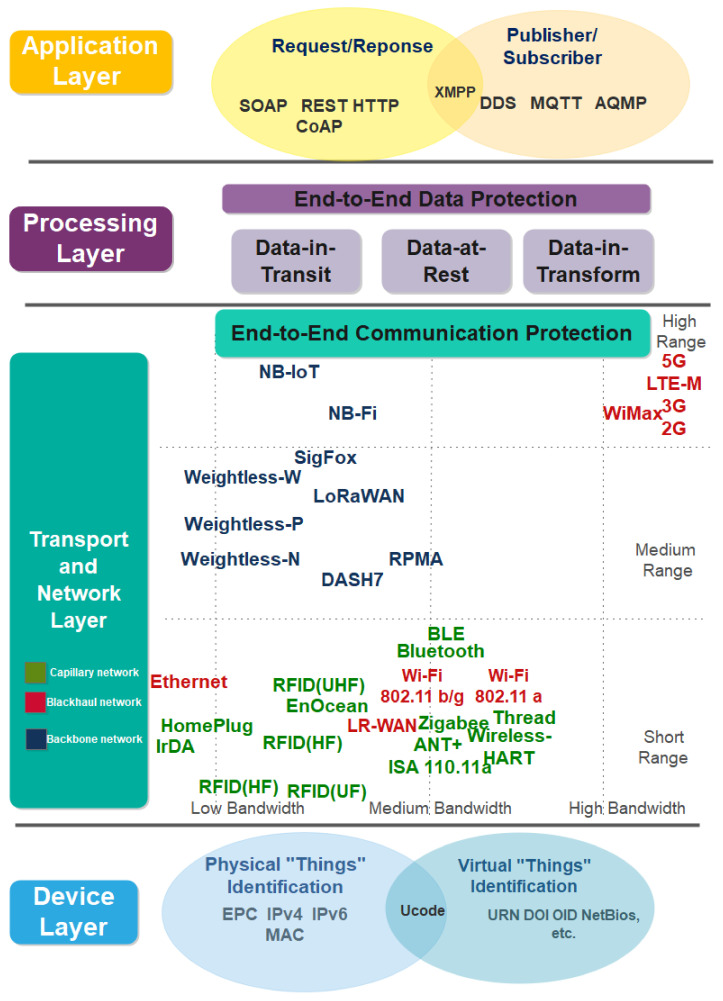
The proposed IIoT security architecture.

**Table 1 sensors-21-06647-t001:** Chronological summary of previous surveys in the IoT and IIoT security.

Year	Reference	S	I	G	O	Focuses
2010	Atzori et al. [1]	√	√		√	Data integrity and privacy issues specifically on wireless technologies: RFID and WSN
	Weber [2]				√	Limited to address data and privacy legislation of the IoT and RFID
2012	Miorandi et al. [3]	√	√		√	A general overview of data confidentiality, privacy and trust specifically on distributed intelligence, communication and identification technologies
2013	Zhao and Ge [4]		√		√	A brief discussion of security attacks and measurements based on three-layer IoT architecture (perception layer, transport layer and application layer)
2014	Ziegeldorf et al. [5]				√	A general overview of IoT privacy threats and challenges
	Jing et al. [6]		√	√	√	Analyze the cross-layer heterogenous and security issues of three-layer IoT architecture (Perception layer, transport layer and application layer) and focuses specifically on WSN and RFID
2015	Fremantle and Scott [7]	√	√		√	Middleware systems and their security properties, as well as a very brief discussion on future works
	Granjal et al. [8]		√		√	IoT communication protocols and technologies specifically on MAC and Physical layers
	Nguyen et al. [9]	√			√	IoT security protocols and key distribution specifically on WSN
2016	Airehrour et al. [10]	√	√		√	Secure routing protocols and trust models
	Qin et al. [11]	√			√	Review IoT from a data-centric perspective, specifically on RFID
2017	Loi et al. [12]	√	√		√	Comprehensive security analysis on consumer IoT Devices
2018	Fernández-Caramés et al. [13]		√		√	Blockchain-based IoT application
2019	Hassija et al. [14]	√			√	Studies on the relationship between IoT application and related technologies: blockchain, machine learning, fog and cloud computing
	Berkay et al. [15]	√			√	Security analysis of IoT programming platforms
	Tabrizi and Pattabiraman [16]	√			√	Design-level and code-level security analysis on IoT devices
2020	Amanullah et al. [17]	√	√		√	Comparative analysis on the relationship of IoT security, deep learning and big data technologies
	Lao et al. [18]	√	√		√	A review on blockchain-based IoT architecture
	Joao et al. [19]	√			√	A general review on threat models and attack path of IoT
2021	Polychronou et al. [20]	√	√			Software attacks targeting hardware vulnerabilities and deep learning detection mechanisms in IIoT
	Gaspar et al. [21]		√		√	A general IoT technologies review on Portugal’s Agro-Industry
	Wu et al. [22]	√			√	Relations between machine learning and blockchain in IIoT
	Latif et al. [23]	√			√	A general review on blockchain-based decentralized IIoT security

Legend: S = security requirements, I = industry countermeasure, G = gap analysis, O = ongoing challenges and future works.

**Table 2 sensors-21-06647-t002:** The difference between conventional systems and IIoT security concerns.

Concerns	Conventional System	IIoT
Connected Nodes/Devices	Small to medium volume within the local networks	Billions of sensor nodes, actuators and automation devices connected
Communication Networks	Homogenous	Heterogeneous
System Scalability	Optional	High scalability The design of IIoT security systems should consider the identification and authentication of an enormous scale of “things”, scalability of communication networks and security key distribution and revocation issues in future
System Interoperability	Optional	High interoperability Diverse security mechanisms and defence systems over the distributed networks must be standardized and compatible with each other to communicate, exchange and process data securely
Collected Data Types	Unified encoding scheme and data format, structured data	Confluent with the terms of “big data” characteristic: High volume (terabytes–zettabytes), High variety (diverse encoding scheme and format, structured data, unstructured data, semi-structured data, quasi-structure data)
Data Processing Model	Moving data to process, moderate speed	Moving processing to data. In most industrial cases, high velocity necessitates real-time analytical processing
Security and Privacy Concerns	Data-at-rest Data-in-memory Data-in-transit	Data-at-rest Data-in-memory Data-in-transit Data-in-transform
Authentication and Access Control Mechanisms	Centralized Approach	Distributed, decentralized approach Lightweight scheme

**Table 3 sensors-21-06647-t003:** CIA+ model of IIoT security.

Security Requirements	Description	IIoT Security Properties
Confidentially(C)	The protection of IIoT from unauthorized disclosure and access.	The security defences and mechanisms should be able to: Protect the connection between “things”, sensing technologies, communication networks Protect data being stored in a data centre or data warehouse (data-at-rest protection) Protect data being transmitted to/from “things”, sensing technologies, communication networks and IIoT applications (data-in-transit protection) Protect the information that being deliver to end-users, such as an analytical result
Integrity(I)	The assurance of IIoT consistency, accuracy, and trustworthiness of data or services over its entire life cycle.	The security mechanisms should be able to detect any data modification and manipulation such as insertion, deletion or replay attacks on the “things” or data of IIoT.
Authentication(A)	The assurance that the communicating entity is the one that it claims to be.	The security mechanisms should be able to ensure: “Things” authentication. Data-origin authentication
Authorization and Access Control (A)	The prevention of unauthorized use of IIoT resources.	The security mechanisms should be able to ensure: Only authorized “things” and users can access IoT networks IIoT edge devices are able to verify whether certain entities are authorized to access their measured data
Availability (A)	The assurance that the IIoT resources are always available.	The security defences and mechanisms should prevent or detect denial of service attacks on IIoT resources.

**Table 4 sensors-21-06647-t004:** Security analysis on electronic product code (EPC) and its object naming service (ONS) architecture.

Security Dimensions	Security Risks and Threats	Deployed Security Countermeasures	Ongoing Challenges and Future Research Works
Confidentiality and Privacy	EPC identifiers and tags data that are stored in “things” with minimal storage and computation power are subjected to identity theft, illegal information disclosure risk and spoofing attack Insecure wireless communication between RFID tags and readers Eavesdroppers capture the incomplete EPC and brute force on the serial number	Secure communication channel via SSL/TLS Privacy protection with routine auto-kill command to destroy EPC information Enforce access password for reading and writing the internal memory of the tag 3DES block cipher and AES hardware cryptographic engine offered by NXP semiconductors [61]	Lightweight data encryption algorithms such as PRESENT block cipher and EPCBC instead of using Kill command Obfuscation algorithm, anonymous mixes approach or onion routing approach to protecting the privacy of EPC (e.g., the source of IP address, the origin of the query) Well-designed network structure (e.g., VPN, Extranets)
Integrity	Malicious attackers controlling intermediate ONS servers and return the forged EPC information Man-in the middle attack on the communication networks between “things” and ONS server Data tampering, loss or corruption of the information stored within the EPC tag	SSL/TLS 192-bit hash function on EPC Gen2 [62,63]	Lightweight hashing algorithms such as PRESENT, SPONGENT, SRFID
Authentication	No mutual authentication between EPC tags and reader and ONS lookup service	SSL/TLS 6-bit pseudo-random number generator (PRNG) and cyclic redundancy code (CRC) on EPC Gen-2 tags	CRC-16 function subjected to brute force attacks and designed protocol not resilient to desynchronisation attacks [64] Lightweight mutual authentication protocol [65,66]
Authorization and Access Control	EPC code can be scanned by any unauthorized reader, thus leading to skimming attacks, spoofing attacks and tag cloning attacks	Enforce 32-bit PIN for reading/writing the internal memory of the tag, as well as a 32-bit PIN for executing an internal auto-killing routine Access control lists	Lightweight access control mechanisms [67] Fine-grained access control list— identity based or attribute-based access control scheme
Availability	Distributed denial-of-service (DDoS) or smurfing attack on the ONS servers or network connections The availability of ONS and EPC resolution services for finding matching information sources with a large scale of “things” growing rapidly	Distributed ONS server Firewalls, intrusion prevention system (IPS), IPS-based prevention, DoS defence system (DDS) based defence.	Distributed query optimization algorithm, distributed architecture for ONS DNS security extensions (DNSSEC) Scalability and extendibility of EPC and ONS services [68,69]

**Table 5 sensors-21-06647-t005:** Security analysis on ubiquitous code (ucode) and its architecture.

Security Dimensions	Security Risks and Threats	Deployed Security Countermeasures	Ongoing Challenges and Future Research Works
Confidentiality and Privacy	Illegal information disclosure risk, spoofing attack, eavesdropping and sniffing on the communication between ucode tags, resolution and information servers	Utilization of the entity transfer protocol (eTP) encryption standard the rule in the exchange of messages [70,71,72] Identity prevention technology that enables the owners of “things” to control information access [70,71]	Lightweight encryption algorithm and communication protocol for constrained resources of IIoT Tamper-resistant tokens [73]
Integrity	Risk of tampering the ucode tags and stored information Man-in-the-middle attack or replay attack on the communication networks between “things”, resolution server and information server	Enforce the ucode tags security, which includes a function to detect missing or lost data, anti-physical duplication/forgery, tamper-resistant, secure communication with unknown node Enforce secure chip to chip communication and rollback transaction mechanism [70,71]	Implement stronger and standardized cryptographic functions in the tiniest and heterogeneous “things”.
Authentication	Fake “things”, malicious ucode, identity theft	eTRON authentication mechanisms constructed based on public key infrastructure (PKI) [72]	Authentication of mass “things” in high velocity Lightweight public encryption schemes such as Rabin’s scheme, NTRU scheme [74] Distributed key management
Authorization and Access Control	Unauthorized access and retrieve ucode information	eTRON access control list	Scalability of key management and access control list
Availability	Very large numbers of “things” and a high volume of inquiries will be sent to a ucode resolution server. It will become challenging to respond to all requests with a single ucode resolution server.	Hierarchical ucode resolution servers	Distributed ucode resolution servers and information servers

**Table 6 sensors-21-06647-t006:** Security analysis on media access control (MAC) and Internet protocol (IP) addresses.

Security Dimensions	Security Risks, Threats and Ongoing Challenges	Deployed Security Countermeasures	Ongoing Challenges and Future Woks
Confidentiality and Privacy	MAC address is subjected to privacy risks of identity confidentiality and location confidentiality, and it can be used to track individuals’ geolocation Reconnaissance IPv4 (e.g., ping sweep or port scanning) Eavesdropping and sniffing on the communication across the MAC layer and network layer such as port scanning, ping sweeps, wiretapping, snooping attack, packet capturing and sniffing attack	RFC 4303 IP encapsulating security payload (ESP) protocol to provide message content confidentiality; however, with a limited traffic flow confidentiality [76] Employs a wide range of data encryption and padding algorithms includes DES, Triple-DES, RC5, IDEA, CAST, CBC, etc.	Lightweight data encryption and padding algorithms such as CLEFIA, PRINCE, KASUMI, K-Cipher-2, Salsa20, ChaCha20 [77]
Integrity and Authentication	Man-in the middle attack such as spoofed ICMPv6 neighbour advertisement and router advertisement, rogue DHCPv6 server, ARP cache poisoning Session hijacking Trust relationship attacks DNS server spoofing attacks Phishing and pharming attack MAC spoofing, masquerading attack, identity theft	RFC 4302 authentication header (AH) protocol to provide data integrity and authentication of IP packets Authentication and key exchange framework, such as IKE, pre-shared keys, Kerberized Internet, OAKLEY, negotiation of keys (KINK), etc. Trust relation management Hash functions such as SHA-1 and MD5 Message authentication code (MAC) such as HMAC-MD5-96, HMAC-SHA-1-96 Spoofing detection software	Lightweight hash functions such as SPONGENT, DM-PRESENT, SPN-Hash, SipHash, PHOTON Lightweight cyclic redundancy check (CRC) message authentication Scalability of key issuing, distribution and management MAC spoofing algorithm Interoperability of heterogeneous communication protocol [78]
Authorization and Access Control	Rogue devices will be as easy to insert into IPv4, as well as IPv6 dual-stack attack, ARP and DHCP attacks IPv4 ARP attack, IPv6 neighbour discovery attack IPv4 DHCP attack, IPv6 stateless auto-configuration attacks	MAC filtering and access control list to permit or deny network access of physical or virtual “things” Standard of secure neighbor discovery (SEND) IPSec policy, IP packet filtering	Fine-grained access control, such as identity-based, attribute-based access control Identity-based generalized sign-cryption for multi-access [67] Prevent gateway guessing scheme
Availability	IP-based flooding attacks such as SYN flood attacks and ICMP flood attacks MAC flooding attacks such as CAM table overflow attack IPv4-based DoS attacks such as smurfing attacks by sending spoofed ICMP echo requests to the broadcast address IPv6-based DoS attacks such as smurfing attack on multicast address, duplication address detection attack, DHCPv6 attack and fragmentation attack	Port security configuration Firewall, intrusion detection and prevention system	Firewalling issues of IPv6 exposes a higher risk as lots of different extension and headers of IPv6 makes it harder for a firewall to filter correctly

**Table 7 sensors-21-06647-t007:** Security analysis on capillary network communication technologies.

Technology /Standard	Features/ Functionalities	Analysis of Security Requirement				Deployed Security Countermeasures	Ongoing Challenges and Future Woks
		C	I	A	V		
IrDA	Use infrared light in a range of <1 m; Application: remote control, data transfer	×	×	×	×	Physical barriers of penetrating walls make IrDA does not employ any data link level security: all data is transmitted without encryption, no authorization, message integrity protection Assume the security protection on application (software level) Both the infrared link access protocol and infrared link management protocol are used to enable simultaneous handshaking and multiplexing of different data streams	Physical security mechanisms to protect the eavesdropping, DoS Attack and privilege escalation Ubiquitous of IrDA application Limited to connect of IIoT “things” within a short distance
RFID	Radio waves with 125 kHz, 13.56 MHz or 902 to 928 MHz within the range of 1 m; Application: tracking, inventory access	√	√	√	√	Unprotected tag data is vulnerable to eavesdropping, spoofing, DoS attack, side-channel attack, etc. Current solutions for tag data protection includes password-based or physical locking of tag memory, tag duplication prevention Privacy protection includes tags’ kill command, faraday cage, active jamming, RSA selective blocker tag, logical hash lock, pseudonyms tags, RFID mutual authentication protocol	Lightweight hash locks Anonymous identity and authentication protocols such as AFMAP, RWP [83,84] RFID virus detection software
NFC	Radio waves with 13.56 MHz, range less than 30 m; Application: payment system, access control, tracking, assisted living	√	√	√	×	ECMA International established several NFC security standards to ensure secure channel and shared service (ECMA-385), data confidentiality and integrity with AES and ECDH (ECMA-386), data authenticated encryption with 256-bit ECDH key agreement and AES in GCM mode (ECMA-409), mutual authentication mechanisms, either with asymmetric cryptography (ECMA-410) or symmetric cryptographic (ECMA-411)	NFC-SEC-01 still vulnerable for main-in-the-middle attack NFCID-1 allows replay of last delivered message
INSTEON	Radio waves with 902 to 924 MHz, range less than 50 m; Application: home automation, domestics	√	√	√	×	INSTEON provides limited security protection as follows: Device access control via linking control and masking non-linked network traffic in which the only user who possesses a device physically can create a link, thus preventing unauthorized access to control neighbours’ devices INSTEON does not support encryption directly, however, via an extended message that contains encrypted payloads in specific applications such as door lock or security systems, includes rolling-code, managed-key, and public-key algorithms, AES-256	Current security mechanisms are less efficient against recent network penetration attacks (e.g., eavesdropping attack by guessing device address). Therefore, more robust access control and authorization mechanisms and lightweight data encryption are needed instead relies on extended message payloads
Bluetooth	Radio waves with 2.4 GHz Medium within range of 10 m and up to 100 m with a higher power; Application: wireless headsets, audio apps, health, animal tagging, intelligent transport systems, Smart home, automotive	√	√	√	×	Enforce access control via symmetric link key that derived from the user entered PIN Enforce device authentication enforced via shared-key challenge/response Bluetooth v2.1 enforce secure simple pairing (SSP) that uses ECDH private key. Enforce confidentiality with the use of SAFER+ algorithm	Short password-based security is vulnerable to password guessing attacks, randomness PIN, scalability of PIN management [84,85,86]
Bluetooth Smart/BLE	Radio waves with 2.4 GHz Medium, Range: >100 m, Application: wearables, gaming, healthcare, sport and fitness	√	√	√	×	Enjoys similar security protection as Bluetooth technologies; however, it enforces stronger security features as follows: 128-bit AES with counter mode CBC-MAC (AES-CCM) Private addressing and data signing via identity resolving key (IRK) and connection signature resolving key (CSRK)	Do not implement end-to-end security and still vulnerable to pairing eavesdropping, man-in-the-middle attacks, DoS attack, fuzzing attack, SSP attack, bluesnarfing, bluebugging, bluejacking [85,86,87]
EnOcean Energy harvesting wireless	Radio Waves with 902.875, 928.35 MHz, 868 MHz, 315 MH within a range up to 30 m (inside buildings) and 300 m (open-air); Application: building automation, transportation, smart home, domestics	√	√	√	×	Enforce confidentiality with 128-bit AES or combine with rolling code for more robust security (variable AES, VAES). Enforce authentication with a combination of 16-bit or 24-bit rolling code, 24-bit or 36-bit CMAC with 128-bit AES. Uses 8-bit checksum to ensure data integrity in the sub-telegram data unit	Security mechanisms for ensuring availability of service and preventing DoS attack, lightweight encryption algorithm and authentication mechanisms [88,89]
Ultra-wideband (UWB) low power and high speed data	Radio waves with 3.1 MHz to 10.6 GHz, Range: <10 m, Application: target detection and tracking, precision navigation, search and rescue, geographic routing, security surveillance, automotive	√	√	√	√	Enforce authentication mechanisms via secure pairing (physical link, visual match confirmation, NFC radio) Enjoys stronger security properties due to its high data rate, low average radiated power, narrow pulses, and very low interference with traditional wireless technologies. It is difficult to conduct wireless network attacks such as jamming and replay attacks Sybil attacks, etc. Subsequently, researchers were adapted UWB technologies to secure RFID and WSN security problems.	Secure positioning algorithms such as pseudo-random turnaround delay protocol, secure localization and authentication algorithm such as SLS, secure device pairing algorithm [90]
ANT+	Radio waves with 2.4 GHz, Range: <10 m; Application: Health and Sport & Fitness Monitoring, Intelligent Transport System, Assisted Live	√	√	×	×	Enforce confidentiality optionally with a 64-bit network key (used to initiate a channel, however inapplicable for encrypting message sent within the channel) and 128-bit AES-CTR algorithm [91]. Uses checksum to verify message content and integrity	AES encryption cannot be used in multichannel mode, forcing the usage of single-channel communications.
HomePlug	Power link communication; Application: smart home, home automation and control and electric vehicle communication applications	√	√	√	×	Enforce confidentiality, authentication and integrity with 128-bit AES encryption derived from user-entered network password	Detected network security vulnerabilities, subjected to remote attacks under default security and authentication settings, lightweight and stronger security mechanisms [92,93]
ZigBee	Radio Waves with 2.4 GHz, Range: <10 m; Application: home monitoring and control, security, smart applications, intelligent transport system, animal tagging, positioning and tracking	√	√	√	×	Employs IEEE 802.15.4 defined security services in Physical and MAC layer and additionally defines its own security model and set of security services at the network and application layers, includes 128-bits key, AES encryption standard (ASE-CCM), Zigbee Trust Center, link key for end-to-end communication and source node authentication, network key for network access control [91]	Still vulnerable to RF-based attacks such as frequency jamming attacks, stronger and lightweight security mechanisms [88,94]
ISA110.11a	Radio waves with 2.4 GHz, Range: <10 m; Application: industrial monitoring and control	√	√	√	×	Enforce confidentiality, integrity and authentication with a set of security keys, includes session key and the network key for secure device-to-device communication, join key (optional) for device authentication Support both symmetric AES-128 and asymmetric (optional) keys for the join process	Password-based schemes such as lightweight hash function enforce availability into security mechanisms, lightweight and stronger encryption algorithm
Wireless-HART	Radio waves with 2.4 GHz, Range: <10 m; Application: industrial monitoring and control	√	√	√	×	Enforce confidentiality, integrity and authentication with a set of security keys, includes session key and the network key for secure device-to-device communication, join key for device authentication Support only symmetric AES-128 key for the join process	Security mechanisms for ensuring availability of service and preventing DoS attack, lightweight encryption algorithm and authentication mechanisms
Thread	Radio waves with 2.4 GHz, Range: <10 m; Application: smart home, building, domestics	√	√	√	×	Follow IEEE 802.15.4 security standards for physical and MAC layer [91] Employ Elliptic Curve variant of J-PAKE (EC-JPAKE) for authentication and key agreement [95]	End-to-end communication and data protection mechanisms

**Table 8 sensors-21-06647-t008:** Security analysis on backhaul network and communication technologies.

Technology/Standard	Features/ Functionalities	Analysis of Security Requirement				Deployed Industry Countermeasure	Ongoing Challenges and Future Woks
		C	I	A	V		
802.3 Ethernet	Range: up to 100 m, Mobility: Portable Bandwidth: 10 Mbps to 10 Gbps shared	√	√	√	×	Enforce confidentiality, integrity, authentication and access control with the following standards: IEEE 802.1AE-2006 that provides confidentiality, integrity, access control on authorized systems includes AES-128 or optional AES-256, Galois Counter Mode-Advanced Encryption Standard-256 (GCM-AES-256), Internet key exchange (IKE)v2 protocol, Extensible Authentication Protocol (EAP), IEEE802.1X-2010 that specifies port-based network access Control IEEE 802.1AR-2009 specifies secure device identifiers (DevIDs) used by various protocols, including IEEE Std. 802.1X-2010 to associate a device with an authentication credential.	Address the reliability, scalability of bandwidth and network size, redundancy and fast network recovery, interoperability with existing commercial standards, employ stronger and lightweight security mechanisms, access control mechanisms such as identity-based or attribute-based access control.
802.11 WLAN/WiFi	Range: 30 m, Mobility: Portable Frequency: 2.4 GHz (802.11b/g), 5.2 GHz (802.11a) Bandwidth: 11–54 Mbps shared	√	√	√	×	Enforce several security modes to ensure confidentiality, authentication, integrity and access control as follows: WEP (wired equivalent privacy) provides authentication and confidentiality with shared WEP keys that generated with 64 or 128-bit RC4 algorithm 802.1X for network access control and timer authentication WPA (Wi-Fi protected access) that aimed to solve the vulnerabilities of WEP by adding message integrity checks and temporal key integrity protocol (TKIP) that generated based on RC4. WP2 (also known as 802.11i WPA) that replaced RC4 with AES-128 bit and TKIP with cipher block chaining message authentication code protocol (CCMP)	Stronger security mechanisms for persistent advanced threats (APT) Lightweight security framework for WiFi-Halow, which aimed for connecting a large number of devices with a constrained resource (low energy, computing power and storage)
802.15.4 LR-WPAN	Range: <10 m Mobility: portable frequency: 868 MHz, 915 MHz, 2450 MHz Bandwidth: up to 250 Kbit/s.	√	√	√	×	Security Specification offers several options of security suites that fulfil different security requirements as follows. Confidentiality and Frame Encryption: AES-CTR, AES-CCM-32, AES-CCM-64, AES-CCM-128 Frame Integrity: AES-CCM-32, AES-CCM-64, AES-CCM-128, AES-CBC-MAC-32, AES-CBC-MAC-64, AES-CBC-MAC-128 Access Protection: AES-CTR, AES-CCM-32, AES-CCM-64, AES-CCM-128, AES-CBC-MAC-32, AES-CBC-MAC-64, AES-CBC-MAC-128 Authentication: AES-CCM-32, AES-CCM-64, AES-CCM-128, AES-CBC-MAC-32, AES-CBC-MAC-64, AES-CBC-MAC-128	Solve the limitation of IEEE 802.15.4 MAC layer security such as performance and reliability issues, jamming attacks on channel hopping Propose security solutions for the newly established IEEE 802.154e standard (time-slotted channel hopping (TSCH) mode
802.16 WiMax	Range: 30 km–50 km, Mobility: Fixed (Mobile -802.16e-2005), Frequency: 2–11 GHz and 23.5–43.5 GHz (802.16a), Bandwidth: up to 70 Mbps shared	√	√	√	×	Enforce confidentiality, integrity, authentication and access control with the IEEE 802.16-2004, IEEE 802.16e-2005, IEEE 802.16-2009, IEEE 802.16j-2009 that provides: - Confidentiality with DES-CBC (IEEE 802.16-2004, IEEE802.16e-2005, IEEE 802.16-2009), AES-CCM/AES-CTR/AES/CBC (IEEE 802.16e-2005, IEEE 802.16-2009) Integrity: AES-CCM (IEEE 802.16e-2005, IEEE 802.16-2009) Authentication and authorization: X.509 certificate, extensible authentication protocol (EAP), PKM (privacy key management) protocol Centralized or distributed multi-hop relay security architecture (IEEE 802.16j-2009)	IEEE 802.16 standards do not address wireless management messages’ availability and confidentiality protection, and end-to-end security is impossible without additional security mechanisms. Scalability of WiMax security architecture to support the dramatic growth of network nodes or “things”
UMTS 3G	Range: UMTS coverage Mobility: full mobility, Frequency: UMTS frequency bands varies on countries (e.g., 2100 MHz for China and Asia, 1900 MHz for US) Bandwidth: 384 Kbps–2 Mbps	√	√	√	×	Provides a better security solution to GSM solutions as follows. Confidentiality: 128-bit KASUMI block cipher algorithm (Mode f8) Authentication: user and serving network authentication, user and mobile station with PIN Data integrity: 128-bit KASUMI and 64-bit MAC (mode f9) Secure international mobile subscriber identity (IMSI): use of temporarily IMSI in the serving network. Network-to-network communication secured with IPSec Authentication and key management: The challenge/response protocol is similar to the GSM subscriber authentication and key establishment protocol with additional sequence number-based one-pass protocol for network authentication derived from ISO/IEC 9798-4.	IMSI is sent without encryption during the first-time user registration with the serving network Hijacking outgoing/incoming calls, man-in-the-man-in-the-middle attacks Security weakness in IPSec

**Table 9 sensors-21-06647-t009:** Security analysis on backbone network and communication technologies.

Technology/Standard	Features/ Functionalities	Analysis of Security Requirement				Deployed Security Countermeasures	Ongoing Challenges and Future Research Woks
		C	I	A	V		
NB-IoT	Radio Waves: 1.4 MHz, 20 MHz,180 kHz Range: Application: smart home, smart city, automotive, energy and logistic	√	√	√	×	Inherits some of the existing LTE security features and uses a partial ciphering mechanism to ensure user data security. Confidentiality: 128-bit EPS encryption algorithms (EEA) (Snow 3G, AES, ZUA), Authentication and access control: EPS AKA procedure Integrity: AS Security keys (e.g., K_eNB_, K_RRCenc_, K_UPenc_), NAS 128-bit integrity algorithm, EPS Integrity Algorithm (EIA) (SNOW 3G, AES, ZUA), 128-bit integrity key (K_NASint_)	Lightweight EEA and EIA mechanisms [106]
Link Labs LoRaWAN	Radio waves: 868 MHz and 915 MHz Range: 7.2 km Network topology: star on star Application: smart cities, smart home	√	√	√	×	Confidentiality: 128-bit AES CTR Integrity: Message Integrity Code (MIC) Relies on network security (TCP/IP SSL) and device encryption keys for authentication Access control and authorization are implemented in the application layer	IETF proposed adding RADIUS protocol for support Authentication, Authorization and Accounting (AAA) (BCP 78 and BCP 79) Stronger data encryption schemes such as AES-CCM-128 End-to-end protection
WAVIoT NB-Fi (cloud-based)	Radio Waves: 915 MHz, 868 MHz, 500 MHz and 433 MHz Range: Up to 16 km in the city and 50 km in the countryside Application: Water Metering, Smart Grid control, parking, smart home, security surveillance	√	-	-	-	Employ symmetric cipher eXtended Tiny Encryption Algorithm (XTEA) to ensure data confidentiality with the length of 256 key. Confidentiality: 256-bit AES CTR and OMAC Integrity: Authenticated encryption with associated data (AEAD) Authentication and access control: network security relies on TCP/IP SSL and VPN technologies	XTEA algorithm is subjected to several differential attacks [107]
NWAVE Weightless-P (Weightless-N, Weightless-W)	Radio Waves: 169 MHz, 433 MHz, 470 MHz, 780 MHz, 868 MHz, 915 MHz, 923 MHz Range: 2 km–5 km Network Topology: Star Application: automotive, sensors, asset tracking, healthcare	√	×	√	×	Password-based authentication and access control mechanisms, data encryption algorithm with AES-128 or AES-256, nonce for preventing replay attack, use temporary device identifiers for privacy protection	Short-password based security mechanism
SigFox UNB Cloud-based	Radio Wave:868 MHz (Europe, Middle East), 902 MHz (North America), 920 MHz (South America, Australia, New Zealand), Range: 9.5 km, Application: Smart Cities, Asset Management, Water Metering, Healthcare, pet tracking, climate monitoring	√	√	√	×	Unique device ID for ensuring identification and authentication via AES encrypted signature, use the sequence number to prevent spoofing attack on transmitted message, data encryption with AES-128	Stronger data encryption scheme such as AES-CCM-128 End-to-end protection
Ingenu RPMA	Radio Waves:2.4 GHz Range: 4.6 km Network topology: star application: smart parking, transportation, tracking, smart building	√	√	√	×	Provides message confidentiality with a 256-bit encryption algorithm, uses a 16-byte hash function for message integrity, subsequently meets the FIPS 140-2 Level 2 encryption standards (e.g., tamper-evidence against unauthorized physical access, role-based authentication, etc.)	Availability and assurance mechanism
DASH7	Radio Waves: 915 MHz, 868 MHz, 500 MHz and 433 MHz Range: 2 km Network topology: star, tree, node-to-node, Application: water metering, smart grid control, parking, smart home, security surveillance	√	√	√	×	Employs AES-128-bit shared encryption, subsequently enforce security properties with data link layer security (AES-128 in EAX mode for authentication and confidentiality, 32-bit integrity check, 56-bit Nonce, 32-bit authentication tag), network layer security and application layer security (secure exchange protocol that possibly constructed part of IPSec.	End-to-end protection

**Table 10 sensors-21-06647-t010:** Security analysis on application layer.

Technology/ Standard	Features/ Functionalities	Analysis of Security Requirement				Deployed Security Countermeasures	Ongoing Challenges and Future Woks
		C	I	A	V		
SOAP	Transport Protocol: HTTP (common use), SMTP, TCP, UDP, JMS; Implement QoS; Architecture: Request/response	√	√	√	×	Use web service security (WS-Security) or SSL to ensure confidentiality, integrity, authentication and access control via: Message Security: X.509 certified, XML signature and XML encryption (e.g., RSA, 3DES) with security tokens, timestamps, error handling procedures, Password-based authentication with username token profile (username and password) WS-Security policy, WS-Secure conversation and W-Trust Use Other XML standards, such as security assertion markup language (SAML)—to bridge the gap between different security models, XML key management specification (XKMS)—to implement PKI in an easier way, XML Access control markup language (XACML) to standardize the access control rule	Password-based attacks on the weak chosen password of username token profile Subjected to application-layer attacks such as malicious input, SQL injection Do not support error handling Interoperability issues to support multiple devices
CoAP	Transport Protocol: UDP; Implement QoS; Architecture: request/response	√	√	√	√	Use Datagram Transport Layer Security (DTLS) that inherits some security feature from TLS/SSL, either: Remain: Null or standard stream cipher, Block Cipher, AEAD ciphers such as ECC-GCM or RSA-GCM With some extensions: record layer, record payload protection, MAC, etc.	DTLS is mandatory for CoAP; however, it is not fully optimized for the resource-constrained network. It inherits from TLS/SSL (e.g., heavy communication flow and buffering required for handshake protocol, heavily of X.509 certificate, etc.) Lightweight cryptography for constrained resources of IIoT such as elliptic curve cryptography (ECC), lightweight or partial SSL Countermeasure privacy threat, network attack such as eavesdropping, man-in-the-middle attack, DoS attack,
REST HTTP	Transport Protocol: HTTP; no implement QoS; Architecture: request/response	√	√	√	×	Use Hypertext Transfer Protocol Secure (HTTPS) that leverages the Transport Layer Security/Secure Socket Layer (TLS/SSL) to provide session-oriented security as follows: Confidentiality with symmetric cryptography (e.g., AES, 3DES, RC4, SCH, blowfish, Twofish) Integrity with keyed MAC and hash functions (e.g., SHA-1, MD5, SHA) Basic authentication with asymmetric cryptography (e.g., RSA, DSA, DDS), X.509 certificate, Diffie–Hellman key exchange protocol and authenticated encryption ciphers (e.g., AES in CCM and GCM mode) Limited authorization and access control such as allowing only authenticated access, anonymous read-only, or specific resource access	Vulnerable to weak key attack, padding oracle attack, adaptive attack with the knowledge of next generated initialization vector, cross-site request forgery, DoS Attack, SSL stripping, network spoofing, traffic analysis Broken cryptographic algorithm (e.g., MD5, DES, PKC#1, etc.) Scalability of PKI, key management and distribution issues Lightweight cryptography algorithm Still suffers from the reliability issues
MQTT	Transport Protocol: TCP; Implement QoS; Architecture: Publish/Subscribe	√	√	√	×	TLS/SSL security mechanism as follows: Confidentiality with symmetric cryptography (e.g., AES, 3DES, RC4, SCH, blowfish, Twofish) Integrity with keyed MAC and hash functions (e.g., SHA-1, MD5, SHA) Basic authentication with asymmetric cryptography (e.g., RSA, DSA, DDS), X.509 certificate, Diffie–Hellman key exchange protocol and authenticated encryption ciphers (e.g., AES in CCM and GCM mode) Limited authorization and access control such as allowing only authenticated access, anonymous read-only, or specific resource access	
XMPP	Transport Protocol: TCP; No Implement QoS; Architecture: Publish/subscribe and request/response	√	√	√	×		
AMQP	Transport Protocol: TCP; Implement QoS; Architecture: publish/subscribe	√	√	√	×		
DDS	Transport Protocol: TCP; Implement QoS; Architecture: publish/subscribe	√	√	√	×	Based on DDS Security Version 1.0 -Beta 2: Confidentiality with symmetric cryptography AES-128/AES-256 Integrity with keyed MAC and hash functions (e.g., MD5, SHA-256) Authentication: authentication between participants and establish a shared secret with AES-GCM-GMAC, X.509 Certificate, RSA, PKI-Diffie–Hellman, Authorization and access control with permission token and access control interface	Subjected to tampering and replay attacks, application-layer attacks, network attacks

## Data Availability

The data presented in this study are available on request from the corresponding author. The data are not publicly available due to the data may involve confidential information of our research group.

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
