# Peer review of "Recent Technologies, Security Countermeasure and Ongoing Challenges of Industrial Internet of Things (IIoT): A Survey"

_sensors, 2021, doi:10.3390/s21196647_

Round 1

Reviewer 1 Report

In general, the manuscript title suggests a "survey" on IIoT "Recent Technologies, Security Countermeasure, and Ongoing Challenges", but it fails to provide a comprehensive survey on the mentioned area. There are several recent papers in this area published in prestigious journals that are not included such as https://doi.org/10.1109/ACCESS.2021.3108130 just to mention a few. The citation in the paper is also cut off apparently and is only appearing 46 of them, while in the paper 63 papers are cited! 

My detailed comments are as follow:

1- In "The contributions of this article" the authors basically provide a workflow, not the exact contributions. This part must be changed. 

2- Section 2 title is "IIoT characteristics and how it is different from conventional security concerns"! This is not a good section title! Please modify.

3- Table 2, please make it readable! As a general comment, try to change all tables structure to make them more readable. Also, for images try to use vector images or high-quality bitmaps so they can be easily readable. 

4- Section 3 title is "IoT Architecture"! Why suddenly you are talking about IoT and not IIoT?!

I think this manuscript needs to be resubmitted after fixing these issues.

Reviewer 2 Report

This paper summarizes IIoT communication technologies with regard to security, focusing on every individual layer of the proposed 4-layer architecture. Although the content of the paper is relevant, the paper itself needs thourough revision due to the inappropriate presentation.

This paper needs to be thoroughly revised language-wise. Authors are not consistent with abbreviations (e.g. Dos-DoS, IIOT-IIoT), number+unit notations (space between a number and a unit "m" in some places present, elsewhere not), spacing issues are present throughout the text. Some sentences have no structure (e.g. subject missing in line 13: "However, fail to....", 2 verbs (e.g. "These include provides")). Every sentence should be proofread and interpunction double-checked.

Tables, although comprehensive, are hardly readable. Have you considering clustering the security mechanisms and visually representing the mapping of technologies to security mecanisms/protocols? At least, I think tables would be more readable if they were in landscape orientation. In addition, Table 2 seems to be missing the top row.

Authors should include more references. They discuss many security mechanisms and technologies and rarely provide references to the background literature on those technologies/security mechanism for more interested readers (the ones who would like more details than available in this paper).

Authors refer to [56]...[62] in the text, but those references are not listed in the list of references. Starting from [21], references in the text do not truthfully map to references in the list of references (I did not check all, but there clearly are inconsistencies). 

Round 2

Reviewer 1 Report

The authors have addressed my previous comments. I have no more comments nor objections against this manuscript. 

Reviewer 2 Report

My concerns were met for the most part.